# *Listeria monocytogenes* requires cellular respiration for NAD⁺ regeneration and pathogenesis

**Rafael Rivera-Lugo**[1†], **David Deng**[1†], **Andrea Anaya-Sanchez**[2], **Sara Tejedor-Sanz**[3,4], **Eugene Tang**[1], **Valeria M Reyes Ruiz**[5,6], **Hans B Smith**[7], **Denis V Titov**[1,8,9], **John-Demian Sauer**[7], **Eric P Skaar**[5,6], **Caroline M Ajo-Franklin**[3,4], **Daniel A Portnoy**[1,10], **Samuel H Light**[11,12]*

[1]Department of Molecular and Cell Biology, University of California, Berkeley, Berkeley, United States; [2]Graduate Group in Microbiology, University of California, Berkeley, Berkeley, United States; [3]Department of Biosciences, Rice University, Houston, United States; [4]The Molecular Foundry, Lawrence Berkeley National Laboratory, Berkeley, United States; [5]Department of Pathology, Microbiology and Immunology, Vanderbilt University Medical Center, Nashville, United States; [6]Vanderbilt Institute for Infection, Immunology, and Inflammation, Vanderbilt University Medical Center, Nashville, United States; [7]Department of Medical Microbiology and Immunology, University of Wisconsin-Madison, Madison, United States; [8]Department of Nutritional Sciences and Toxicology, University of California, Berkeley, Berkeley, United States; [9]Center for Computational Biology, University of California, Berkeley, Berkeley, United States; [10]Department of Plant and Microbial Biology, University of California, Berkeley, Berkeley, United States; [11]Department of Microbiology, University of Chicago, Chicago, United States; [12]Duchossois Family Institute, University of Chicago, Chicago, United States

**\*For correspondence:**
samlight@uchicago.edu

[†]These authors contributed equally to this work

**Abstract** Cellular respiration is essential for multiple bacterial pathogens and a validated antibiotic target. In addition to driving oxidative phosphorylation, bacterial respiration has a variety of ancillary functions that obscure its contribution to pathogenesis. We find here that the intracellular pathogen *Listeria monocytogenes* encodes two respiratory pathways which are partially functionally redundant and indispensable for pathogenesis. Loss of respiration decreased NAD⁺ regeneration, but this could be specifically reversed by heterologous expression of a water-forming NADH oxidase (NOX). NOX expression fully rescued intracellular growth defects and increased *L. monocytogenes* loads >1000-fold in a mouse infection model. Consistent with NAD⁺ regeneration maintaining *L. monocytogenes* viability and enabling immune evasion, a respiration-deficient strain exhibited elevated bacteriolysis within the host cytosol and NOX expression rescued this phenotype. These studies show that NAD⁺ regeneration represents a major role of *L. monocytogenes* respiration and highlight the nuanced relationship between bacterial metabolism, physiology, and pathogenesis.

## Editor's evaluation

In this study, authors report that a major role of respiration in *Listeria monocytogenes* pathogenicity, is to control redox balance (NAD⁺ regeneration) rather than generation of proton motive force. This is a new way of perceiving respiration that should be of interest to the microbiology community and broader readership.

**eLife digest** Cellular respiration is one of the main ways organisms make energy. It works by linking the oxidation of an electron donor (like sugar) to the reduction of an electron acceptor (like oxygen). Electrons pass between the two molecules along what is known as an 'electron transport chain'. This process generates a force that powers the production of adenosine triphosphate (ATP), a molecule that cells use to store energy.

Respiration is a common way for cells to replenish their energy stores, but it is not the only way. A simpler process that does not require a separate electron acceptor or an electron transport chain is called fermentation. Many bacteria have the capacity to perform both respiration and fermentation and do so in a context-dependent manner.

Research has shown that respiration can contribute to bacterial diseases, like tuberculosis and listeriosis (a disease caused by the foodborne pathogen *Listeria monocytogenes*). Indeed, some antibiotics even target bacterial respiration. Despite being often discussed in the context of generating ATP, respiration is also important for many other cellular processes, including maintaining the balance of reduced and oxidized nicotinamide adenine dinucleotide (NAD) cofactors. Because of these multiple functions, the exact role respiration plays in disease is unknown.

To find out more, Rivera-Lugo, Deng et al. developed strains of the bacterial pathogen *Listeria monocytogenes* that lacked some of the genes used in respiration. The resulting bacteria were still able to produce energy, but they became much worse at infecting mammalian cells. The use of a genetic tool that restored the balance of reduced and oxidized NAD cofactors revived the ability of respiration-deficient *L. monocytogenes* to infect mammalian cells, indicating that this balance is what the bacterium requires to infect.

Research into respiration tends to focus on its role in generating ATP. But these results show that for some bacteria, this might not be the most important part of the process. Understanding the other roles of respiration could change the way that researchers develop antibacterial drugs in the future. This in turn could help with the growing problem of antibiotic resistance.

## Introduction

Distinct metabolic strategies allow microbes to extract energy from diverse surroundings and colonize nearly every part of the earth. Microbial energy metabolisms vary greatly but can be generally categorized as possessing fermentative or respiratory properties. Cellular respiration is classically described by a multistep process that initiates with the enzymatic oxidation of organic matter and the accompanying reduction of $NAD^+$ (nicotinamide adenine dinucleotide) to NADH. Respiration of fermentable sugars typically starts with glycolysis, which generates pyruvate and NADH. Pyruvate then enters the tricarboxylic acid (TCA) cycle, where its oxidation to carbon dioxide is coupled to the production of additional NADH. NADH generated by glycolysis and the TCA cycle is then oxidized by NADH dehydrogenase to regenerate $NAD^+$ and the resulting electrons are transferred via an electron transport chain to a terminal electron acceptor.

While mammals strictly use oxygen as a respiratory electron acceptor, microbes reside in diverse oxygen-limited environments and have varying and diverse capabilities to use disparate non-oxygen respiratory electron acceptors. Whatever the electron acceptor, electron transfer in the electron transport chain is often coupled to proton pumping across the bacterial inner membrane. This generates a proton gradient or proton motive force, which powers a variety of processes, including ATP production by ATP synthase.

Respiratory pathways are important for several aspects of bacterial physiology. Respiration's role in establishing the proton motive force allows bacteria to generate ATP from non-fermentable energy sources (which are not amenable to ATP production by substrate-level phosphorylation) and increases ATP yields from fermentable energy sources. In addition to these roles in ATP production, respiratory electron transport chains are directly involved in many other aspects of bacterial physiology, including the regulation of cytosolic pH, transmembrane solute transport, ferredoxin-dependent metabolisms, protein secretion, protein folding, disulfide formation, and flagellar motility (*Bader et al., 1999*; *Driessen et al., 2000*; *Driessen and Nouwen, 2008*; *Manson et al., 1977*; *Slonczewski et al., 2009*; *Driessen et al., 2000*; *Tremblay et al., 2013*; *Wilharm et al., 2004*). Beyond the proton motive force,

respiration functions to regenerate $NAD^+$, which is essential for enabling the continued function of glycolysis and other metabolic processes. By obviating fermentative mechanisms of $NAD^+$ regeneration, respiration increases metabolic flexibility, which, among other metabolic consequences, can enhance ATP production by substrate-level phosphorylation (*Hunt et al., 2010*).

Bacterial pathogens reside within a host where they must employ fermentative or respiratory metabolisms to power growth. Pathogen respiratory processes have been linked to host-pathogen conflict in several contexts. Phagocytic cells target bacteria by producing reactive nitrogen species that inhibit aerobic respiration (*Richardson et al., 2008*). *Aggregatibacter actinomycetemcomitans*, *Salmonella enterica*, *Streptococcus agalactiae,* and *Staphylococcus aureus* mutants with impaired aerobic respiration are attenuated in murine models of systemic disease (*Craig et al., 2013*; *Hammer et al., 2013*; *Jones-Carson et al., 2016*; *Lencina et al., 2018*; *Lewin et al., 2019*; *Rivera-Chávez et al., 2016*). Aerobic respiration is vital for *Mycobacterium tuberculosis* pathogenesis and persister cell survival, making respiratory systems validated anti-tuberculosis drug targets (*Cook et al., 2014*; *Hasenoehrl et al., 2020*). Respiratory processes that use oxygen, tetrathionate, and nitrate as electron acceptors

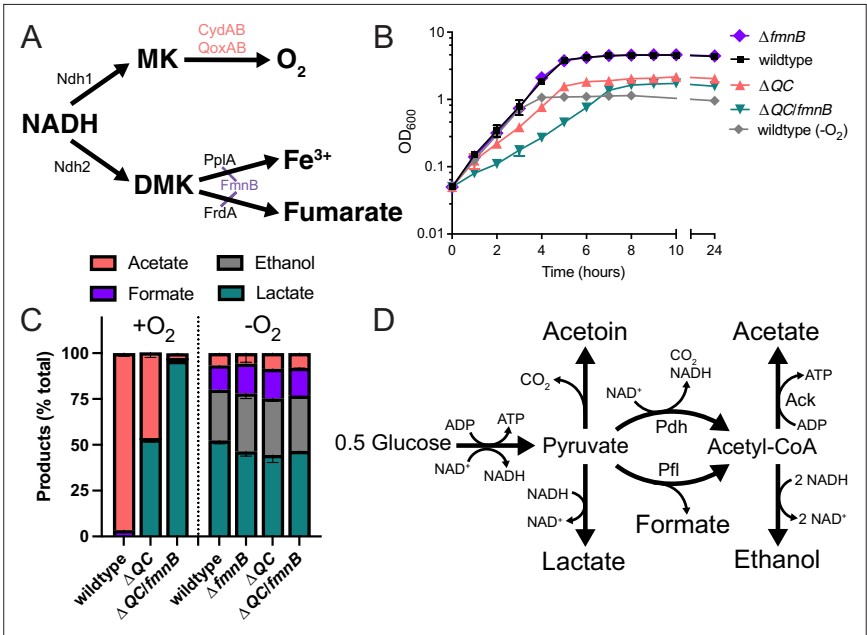

**Figure 1.** Respiration impacts *L. monocytogenes* growth and fermentative output. (**A**) Proposed respiratory electron transport chains in *L. monocytogenes*. Different NADH dehydrogenases likely transfer electrons to distinct but presently unidentified quinones ($Q_a$ and $Q_b$). FmnB catalyzes assembly of essential components of the electron transport chain, PplA and FrdA, that can transfer electrons to ferric iron and fumarate, respectively. Other proteins involved in the terminal electron transfer steps are noted. (**B**) Optical density of *L. monocytogenes* strains aerobically grown in nutrient-rich media, with the anaerobically grown wildtype strain provided for context. The means and standard deviations from three independent experiments are shown. (**C**) Fermentation products of *L. monocytogenes* strains grown to stationary phase in nutrient-rich media under aerobic and anaerobic conditions. Error bars show standard deviations. Results from three independent experiments are shown. (**D**) Proposed pathways for *L. monocytogenes* sugar metabolism. The predicted number of NADH generated (+) or consumed (−) in each step is indicated. PplA, peptide pheromone-encoding lipoprotein A; FrdA, fumarate reductase; ΔQC, ΔqoxA/ΔcydAB; ΔQC/fmnB, ΔqoxA/ΔcydAB/fmnB::tn; GLC, glucose; Ack, acetate kinase; Pdh, pyruvate dehydrogenase; Pfl, pyruvate formate-lyase; DMK, demethylmenaquinone.

The online version of this article includes the following source data and figure supplement(s) for figure 1:

**Source data 1.** Source data for *Figure 1B*.

**Source data 2.** Source data for *Figure 1C*.

**Figure supplement 1.** Use of respiratory electron acceptors enhances *Listeria monocytogenes* growth in nutrient-rich media.

**Figure supplement 1—source data 1.** Source data for *Figure 1—figure supplement 1A*.

**Figure supplement 1—source data 2.** Source data for *Figure 1—figure supplement 1B*.

are important for the growth of *S. enterica* and *Escherichia coli* in the mammalian intestinal lumen (*Rivera-Chávez et al., 2016*; *Winter et al., 2010*; *Winter et al., 2013*). While several studies have linked respiration in bacterial pathogens to the use of specific electron donors (i.e. non-fermentable energy sources) within the intestinal lumen, the particular respiratory functions important for systemic bacterial infections remain largely unexplained (*Ali et al., 2014*; *Faber et al., 2017*; *Gillis et al., 2018*; *Spiga et al., 2017*; *Thiennimitr et al., 2011*).

*Listeria monocytogenes* is a human pathogen that, after being ingested on contaminated food, can gain access to the host cell cytosol and use actin-based motility to spread from cell to cell (*Freitag et al., 2009*). *L. monocytogenes* has two respiratory-like electron transport chains. One electron transport chain is dedicated to aerobic respiration and uses a menaquinone intermediate and QoxAB ($aa_3$) or CydAB (*bd*) cytochrome oxidases for terminal electron transfer to $O_2$ (*Figure 1A*; *Corbett et al., 2017*). We recently identified a second flavin-based electron transport chain that transfers electrons to extracytosolic acceptors (including ferric iron and fumarate) via a putative demethylmenaquinone intermediate and can promote growth in anaerobic conditions (*Figure 1A*; *Light et al., 2018*; *Light et al., 2019*; *Zeng et al., 2021*). Final electron transfer steps in this flavin-based electron transport mechanism are catalyzed by PplA and FrdA, which are post-translationally linked to an essential cofactor by the flavin mononucleotide transferase (FmnB) (*Light et al., 2018*; *Méheust et al., 2021*).

*L. monocytogenes* resembles fermentative microbes in lacking a functional TCA cycle (*Trivett and Meyer, 1971*). Despite thus being unable to completely oxidize sugar substrates, previous studies have shown that aerobic respiration is important for the systemic spread of *L. monocytogenes* (*Chen et al., 2017*; *Corbett et al., 2017*; *Stritzker et al., 2004*). Microbes that similarly contain a respiratory electron transport chain but lack a TCA cycle are considered to employ a respiro-fermentative metabolism (*Pedersen et al., 2012*). Respiro-fermentative metabolisms tune the cell's fermentative output and often manifest with the respiratory regeneration of $NAD^+$ enabling a shift from the production of reduced (e.g. lactic acid and ethanol) to oxidized (e.g. acetic acid) fermentation products. In respiro-fermentative lactic acid bacteria closely related to *L. monocytogenes*, cellular respiration results in a modest growth enhancement, but is generally dispensable (*Duwat et al., 2001*; *Pedersen et al., 2012*).

The studies presented here sought to address the role of respiration in *L. monocytogenes* pathogenesis. Our results confirm that *L. monocytogenes* exhibits a respiro-fermentative metabolism and show that its two respiratory systems are partially functionally redundant under aerobic conditions. We find that the respiration-deficient *L. monocytogenes* strains exhibit severely attenuated virulence and lyse within the cytosol of infected cells. Finally, we selectively abrogate the effect of diminished $NAD^+$ regeneration in respiration-deficient *L. monocytogenes* strains by heterologous expression of a water-forming NADH oxidase (NOX) and find that this restores virulence. These results thus elucidate the basis of *L. monocytogenes* cellular respiration and demonstrate that $NAD^+$ regeneration represents a key function of this activity in *L. monocytogenes* pathogenesis.

## Results

### *L. monocytogenes'* electron transport chains have distinct roles in aerobic and anaerobic growth

We selected previously characterized Δ*qoxA*/Δ*cydAB* (ΔQC) and Δ*fmnB L. monocytogenes* strains to study the role of aerobic respiration and extracellular electron transfer, respectively (*Chen et al., 2017*; *Light et al., 2018*). In addition, we generated a Δ*qoxA*/Δ*cydAB*/*fmnB*::tn (ΔQC/*fmnB*) *L. monocytogenes* strain to test for functional redundancies of aerobic respiration and extracellular electron transfer. Initial studies measured the growth of these strains on nutritionally rich brain heart infusion (BHI) media in the presence/absence of electron acceptors.

Compared to anaerobic conditions that lacked an electron acceptor, we found that aeration led to a relatively modest increase in growth of wildtype and Δ*fmnB* strains (*Figure 1B* and *Figure 1—figure supplement 1a*). This growth enhancement could be attributed to aerobic respiration, as aerobic growth of the ΔQC strain resembled anaerobically cultured strains (*Figure 1B* and *Figure 1—figure supplement 1a*). Similarly, in anaerobic conditions, inclusion of the extracellular electron acceptors, ferric iron and fumarate, resulted in a small growth enhancement of wildtype *L. monocytogenes* (*Figure 1—figure supplement 1b*). This phenotype could be attributed to extracellular electron

transfer, as ferric iron or fumarate failed to stimulate growth of the ΔfmnB strain (*Figure 1—figure supplement 1b*). These findings are consistent with aerobic respiration and extracellular electron transfer possessing distinct roles in aerobic and anaerobic environments, respectively.

The ΔQC/fmnB strain exhibited the most striking growth pattern. This strain lacked a phenotype under anaerobic conditions but had impaired aerobic growth, even relative to the ΔQC strain (*Figure 1B*). Notably, ΔQC/fmnB was the sole strain tested with a substantially reduced growth rate in the presence of oxygen (*Figure 1B*). These observations suggest that aerobic extracellular electron transfer activity can partially compensate for the loss of aerobic respiration and that oxygen inhibits *L. monocytogenes* growth in the absence of both electron transport chains.

## Respiration alters *L. monocytogenes'* fermentative output

Respiration is classically defined by the complete oxidation of an electron donor (e.g. glucose) to carbon dioxide in the TCA cycle. However, *L. monocytogenes* lacks a TCA cycle and instead converts sugars into multiple fermentation products (*Romick et al., 1996*). We thus asked how respiration impacts *L. monocytogenes'* fermentative output. Under anaerobic conditions that lacked an alternative electron acceptor, *L. monocytogenes* exhibited a pattern of mixed acid fermentation, with lactic acid being most abundant and ethanol, formic acid, and acetic acid being produced at lower levels (*Figure 1C*). By contrast, under aerobic conditions *L. monocytogenes* almost exclusively produced acetic acid (*Figure 1C*). Consistent with respiration being partially responsible for the distinct aerobic vs. anaerobic responses, ΔQC and ΔQC/fmnB strains failed to undergo drastic shifts in fermentative output when grown in aerobic conditions. The ΔQC strain mainly produced lactic acid in the presence of oxygen, and this trend was even more pronounced in the ΔQC/fmnB strain, which almost exclusively produced lactic acid (*Figure 1C*). These results show that aerobic respiration induces a shift to acetic acid production and support the conclusion that *L. monocytogenes'* two electron transport chains are partially functionally redundant in aerobic conditions.

A comparison of fermentative outputs across the experimental conditions also clarifies the basis of central energy metabolism in *L. monocytogenes*. A classical glycolytic metabolism in *L. monocytogenes* likely generates ATP and NADH. In the absence of oxygen or an alternative electron acceptor,

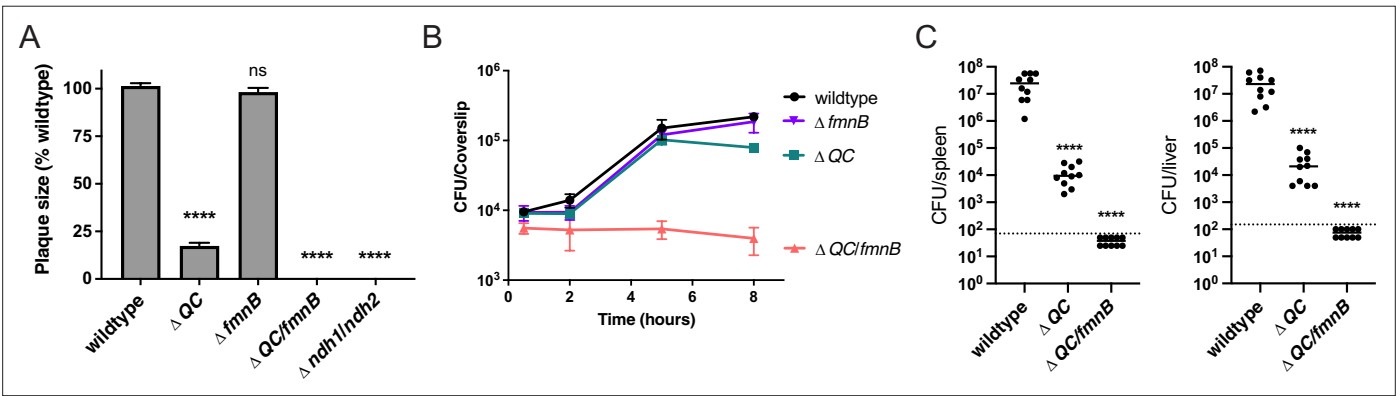

**Figure 2.** Respiration is required for *L. monocytogenes* virulence. (**A**) Plaque formation by cell-to-cell spread of *L. monocytogenes* strains in monolayers of mouse L2 fibroblast cells. The mean plaque size of each strain is shown as a percentage relative to the wildtype plaque size. Error bars represent standard deviations of the mean plaque size from two independent experiments. Statistical analysis was performed using one-way ANOVA and Dunnett's post-test comparing wildtype to all the other strains. ****, p<0.0001; ns, no significant difference (p>0.05). (**B**) Intracellular growth of *L. monocytogenes* strains in murine bone marrow-derived macrophages (BMMs). At 1-hour post-infection, infected BMMs were treated with 50 μg/mL of gentamicin to kill extracellular bacteria. Colony-forming units (CFU) were enumerated at the indicated times. Results are representative of two independent experiments. (**C**) Bacterial burdens in murine spleens and livers 48 hours post-intravenous infection with indicated *L. monocytogenes* strains. The median values of the CFUs are denoted by black bars. The dashed lines represent the limit of detection. Data were combined from two independent experiments, *n* = 10 mice per strain. Statistical significance was evaluated using one-way ANOVA and Dunnett's post-test using wildtype as the control. ****, p<0.0001. ΔQC, ΔqoxA/ΔcydAB; ΔQC/fmnB, ΔqoxA/ΔcydAB/fmnB::tn; Δndh1/ndh2, Δndh1/ndh2::tn.

The online version of this article includes the following source data for figure 2:

**Source data 1.** Source data for *Figure 2A*.

**Source data 2.** Source data for *Figure 2B*.

**Source data 3.** Source data for *Figure 2C*.

NAD[+] is regenerated by coupling NADH oxidation to the reduction of pyruvate to lactate or ethanol. In the presence of oxygen, NADH oxidation is coupled to the reduction of oxygen, and pyruvate is converted to acetate. Moreover, the pattern of anaerobic formate production is consistent with aerobic acetyl-CoA production through pyruvate dehydrogenase and anaerobic production through pyruvate formate-lyase (*Figure 1D*). Collectively, these observations suggest that *L. monocytogenes* prioritizes balancing NAD[+]/NADH levels in the absence of an electron acceptor and maximizing ATP production in the presence of oxygen. In the absence of oxygen, NAD[+]/NADH redox homeostasis is achieved by minimizing NADH produced in acetyl-CoA biosynthesis and by consuming NADH in lactate/ethanol fermentation (*Figure 1D*). In the presence of oxygen, ATP yields are maximized through respiration and increased substrate-level phosphorylation by acetate kinase activity (*Figure 1D*).

## Respiratory capabilities are essential for *L. monocytogenes* pathogenesis

We next asked about the role of cellular respiration in intracellular *L. monocytogenes* growth and pathogenesis. The Δ*fmnB* mutant deficient for extracellular electron transfer was previously shown to resemble the wildtype *L. monocytogenes* strain in a murine model of infection (*Light et al., 2018*). We found that this mutant also did not differ from wildtype *L. monocytogenes* in growth in bone marrow-derived macrophages (BMMs) and a plaque assay that monitors bacterial growth and cell-to-cell spread (*Figure 2A and B*). Consistent with previous reports, the Δ*QC* strain deficient for aerobic respiration was attenuated in the plaque assay and murine model of infection, but resembled wildtype *L. monocytogenes* in macrophage growth (*Figure 2A–C*; *Chen et al., 2017*; *Corbett et al., 2017*). Combining mutations that resulted in the loss of both extracellular electron transfer and aerobic respiration produced even more pronounced phenotypes. The Δ*QC/fmnB* strain did not grow

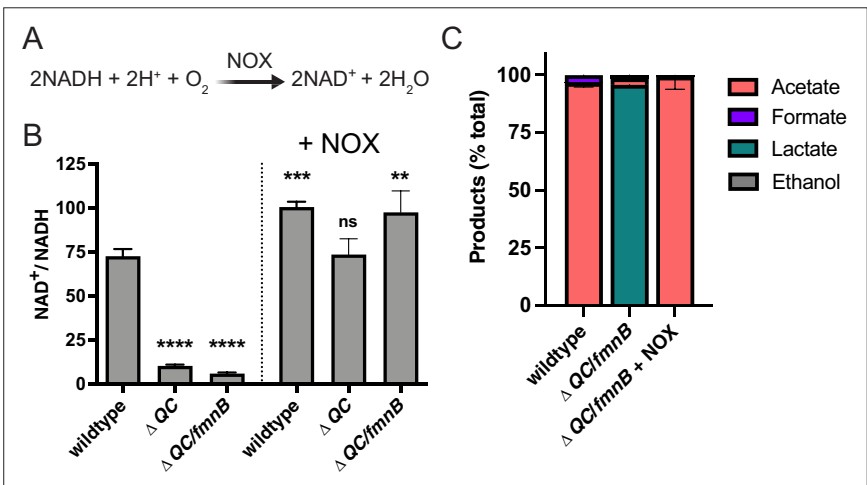

**Figure 3.** Water-forming NADH oxidase (NOX) restores redox homeostasis in respiration-deficient *L. monocytogenes* strains. (**A**) Reaction catalyzed by the *Lactococcus lactis* water-forming NOX, which is the same as aerobic respiration without the generation of a proton motive force. (**B**) NAD[+]/NADH ratios of parent and NOX-complemented *L. monocytogenes* strains grown aerobically in nutrient-rich media to mid-logarithmic phase. Results from three independent experiments are presented as means and standard deviations. Statistical significance was calculated using one-way ANOVA and Dunnett's post-test using the wildtype parent strain as the control. ****, $p < 0.0001$; ***, $p < 0.001$; **, $p < 0.01$; ns, not statistically significant ($p > 0.05$). (**C**) Fermentation products of *L. monocytogenes* strains grown in nutrient-rich media under aerobic conditions. Error bars show standard deviations. Results from three independent experiments are shown. Δ*QC*, Δ*qoxA*/Δ*cydAB*; Δ*QC/fmnB*, Δ*qoxA*/Δ*cydAB*/*fmnB*::tn; + NOX, strains complemented with *L. lactis nox*.

The online version of this article includes the following source data and figure supplement(s) for figure 3:

**Source data 1.** Source data for *Figure 3B*.

**Source data 2.** Source data for *Figure 3C*.

**Figure supplement 1.** NOX expression in respiration-deficient mutants fails to rescue swarming motility.

**Figure supplement 1—source data 1.** Source data for *Figure 3—figure supplement 1*.

intracellularly in macrophages and fell below the limit of detection in the plaque assay and murine infection model (*Figure 2A–C*). Consistent with this phenotype reflecting a loss of respiratory activity, we observed that a mutant that targeted the two respiratory NADH dehydrogenases resulted in a similar phenotype in the plaque assay (*Figure 2A*). These results thus demonstrate that respiratory activities are essential for *L. monocytogenes* virulence, and that the organism's two respiratory pathways are partially functionally redundant within a mammalian host.

## Expression of NOX restores NAD$^+$ levels in *L. monocytogenes* respiration mutants

Cellular respiration both regenerates NAD$^+$ and establishes a proton motive force that is important for various aspects of bacterial physiology. The involvement of respiration in these two distinct processes can confound the analysis of respiration-impaired phenotypes. However, the heterologous expression of water-forming NADH oxidase (NOX) has been used to decouple these functionalities in mammalian cells (*Figure 3A*; *Titov et al., 2016*). Because NOX regenerates NAD$^+$ without pumping protons across the membrane, its introduction to a respiration-deficient cell can correct the NAD$^+$/NADH imbalance, thereby isolating the role of the proton motive force in the phenotype (*Lopez de Felipe et al., 1998*; *Titov et al., 2016*).

To address which aspect of cellular respiration was important for *L. monocytogenes* pathogenesis, we introduced the previously characterized *Lactococcus lactis* water-forming NOX to the genome of respiration-deficient *L. monocytogenes* strains (*Heux et al., 2006*; *Neves et al., 2002a*; *Neves et al., 2002b*). We confirmed that the Δ*QC* and Δ*QC/fmnB* strains exhibited decreased NAD$^+$/NADH levels and that constitutive expression of NOX rescued this phenotype (*Figure 3B*). Consistent with the altered fermentative output of the Δ*QC/fmnB* strain resulting from impaired NAD$^+$ regeneration, we observed that NOX expression restored the predominance of acetic acid production to the aerobically grown cells (*Figure 3C*).

To confirm that NOX expression specifically impacts NAD$^+$/NADH-dependent phenotypes, we tested the effect of NOX expression on bacterial motility. Consistent with respiration impacting flagellar function through the proton motive force, we found that Δ*QC/fmnB* exhibited impaired bacterial motility and that this phenotype was resilient to NOX expression (*Manson et al., 1977*; *Figure 3—figure supplement 1*). These experiments thus provide evidence that NOX expression provides a tool to specifically manipulate the NAD$^+$/NADH ratio in *L. monocytogenes*.

## Respiration is critical for regenerating NAD$^+$ during *L. monocytogenes* pathogenesis

We next sought to dissect the relative importance of respiration in generating a proton motive force versus maintaining redox homeostasis for *L. monocytogenes* virulence. We tested NOX-expressing Δ*QC* and Δ*QC/fmnB* strains for macrophage growth, plaque formation, and in the murine infection model. Expression of NOX almost fully rescued the plaque and macrophage growth phenotypes of the Δ*QC* and Δ*QC/fmnB* strains (*Figure 4A and B*). NOX expression also partially rescued *L. monocytogenes* virulence in the murine infection model (*Figure 4C*). Notably, NOX expression had a greater impact on the *L. monocytogenes* load in the spleen than the liver, suggesting distinct functions of respiration for *L. monocytogenes* colonization of these two organs (*Figure 4C*). These results thus suggest that NAD$^+$ regeneration represents the primary role of respiration in *L. monocytogenes* pathogenesis to an organ-specific extent.

## Impaired redox homeostasis is associated with increased cytosolic *L. monocytogenes* lysis

We next asked why respiration-mediated redox homeostasis was critical for *L. monocytogenes* pathogenesis. We reasoned that previous descriptions of *L. monocytogenes* quinone biosynthesis mutants might provide a clue. Quinones are a family of redox-active cofactors that have essential functions in respiratory electron transport chains (*Collins and Jones, 1981*). Our previous studies suggested that distinct quinones function in flavin-based electron transfer and aerobic respiration (*Light et al., 2018*). A separate set of studies found that *L. monocytogenes* quinone biosynthesis mutants exhibited divergent phenotypes. *L. monocytogenes* strains defective in upstream steps of the quinone biosynthesis pathway (*menB*, *menC*, *menD*, *menE*, and *menF*) exhibited increased bacteriolysis in the cytosol of

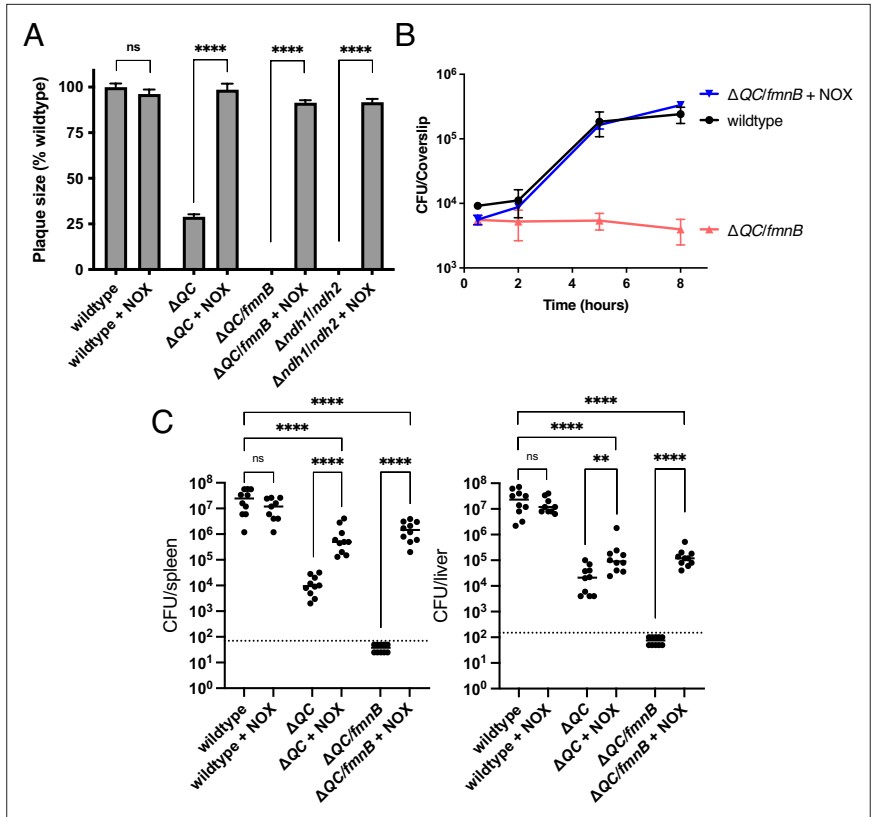

**Figure 4.** NOX expression restores virulence to respiration-deficient *L. monocytogenes* strains. (**A**) Plaque formation by cell-to-cell spread of *L. monocytogenes* strains in monolayers of mouse L2 fibroblast cells. The mean plaque size of each strain is shown as a percentage relative to the wildtype plaque size. Error bars represent standard deviations of the mean plaque size from two independent experiments. Statistical analysis was performed using the unpaired two-tailed *t* test. ****, p<0.0001; ns, no significant difference (p>0.05). (**B**) Intracellular growth of *L. monocytogenes* strains in murine bone marrow-derived macrophages (BMMs). At 1-hour post-infection, infected BMMs were treated with 50 µg/mL of gentamicin to kill extracellular bacteria. Colony-forming units (CFU) were enumerated at the indicated times. Results are representative of three independent experiments. (**C**) Bacterial burdens in murine spleens and livers 48 hours post-intravenous infection with indicated *L. monocytogenes* strains. The median values of the CFUs are denoted by black bars. The dashed lines represent the limit of detection. Data were combined from two independent experiments, *n* = 10 mice per strain, but for the wildtype +NOX strain (*n* = 9 mice). Statistical significance was evaluated using one-way ANOVA and Dunnett's post-test using the wildtype control strain to compare with the NOX-complemented strains. Significance between the parental and the NOX-complemented strains was determined using the unpaired two-tailed *t* test. ****, p<0.0001; **, p<0.01; ns, no significant difference (p>0.05). ΔQC, Δ*qoxA*/Δ*cydAB*; ΔQC/*fmnB*, Δ*qoxA*/Δ*cydAB*/*fmnB*::tn; Δ*ndh1*/*ndh2*, Δ*ndh1*/*ndh2*::tn; + NOX, strains complemented with *Lactococcus lactis nox*.

The online version of this article includes the following source data for figure 4:

**Source data 1.** Source data for *Figure 4A*.

**Source data 2.** Source data for *Figure 4B*.

**Source data 3.** Source data for *Figure 4C*.

host cells and were severely attenuated for virulence (*Figure 5A*). By contrast, *L. monocytogenes* strains defective in downstream steps of the quinone biosynthesis pathway (*menA* and *menG*) did not exhibit increased cytosolic bacteriolysis and had less severe virulence phenotypes (*Chen et al., 2019*, *Chen et al., 2017*; *Smith et al., 2021*; *Figure 5A*). These divergent phenotypic responses resemble the loss of aerobic respiration versus the loss of aerobic respiration *plus* flavin-based electron transfer observed in our studies. The distinct virulence phenotype of quinone biosynthesis mutants could thus be explained by the upstream portion of the quinone biosynthesis pathway being required for both

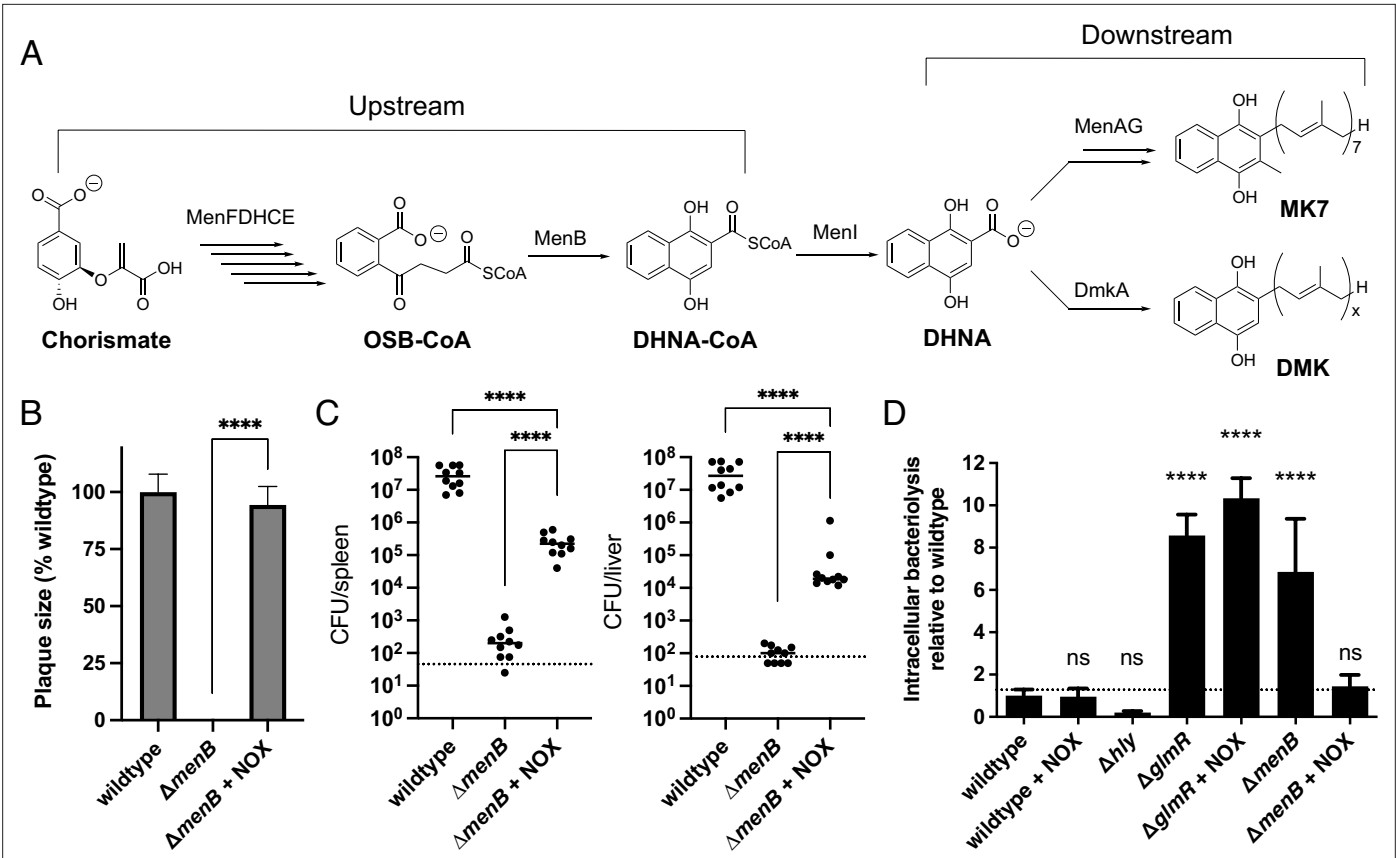

**Figure 5.** Impaired redox homeostasis accounts for elevated bacteriolysis of a respiration-deficient *L. monocytogenes* strain in the cytosol of infected cells. (**A**) Proposed *L. monocytogenes* quinone biosynthesis pathway. Arrows indicate the number of enzymes that catalyze each reaction. An unidentified demethylmenaquinone (DMK) is proposed to be required for the flavin-based electron transfer pathway and MK7 required for aerobic respiration. Loss of the upstream portion of the pathway is anticipated to impact both electron transport chains. (**B**) Plaque formation by cell-to-cell spread of *L. monocytogenes* strains in monolayers of mouse L2 fibroblast cells. The mean plaque size of each strain is shown as a percentage relative to the wildtype plaque size. Error bars represent standard deviations of the mean plaque size from two independent experiments. Statistical analysis was performed using the unpaired two-tailed *t* test. ****, p<0.0001. (**C**) Bacterial burdens in murine spleens and livers 48 hours post-intravenous infection with indicated *L. monocytogenes* strains. The median values of the CFUs are denoted by black bars. The dashed lines represent the limit of detection. Data were combined from two independent experiments, *n* = 10 mice per strain. Statistical significance was evaluated using one-way ANOVA and Dunnett's post-test using the wildtype strain as the control to compare with the NOX-complemented strain. Significance between the parental and the NOX-complemented strain was determined using the unpaired two-tailed *t* test. ****, p<0.0001. (**D**) Bacteriolysis of *L. monocytogenes* strains in bone marrow-derived macrophages. The data are normalized to wildtype bacteriolysis levels and presented as means and standard deviations from three independent experiments. Statistical significance was calculated using one-way ANOVA and Dunnett's post-test using the wildtype parent strain as the control. ****, p<0.0001; ns, no significant difference (p>0.05).

The online version of this article includes the following source data for figure 5:

**Source data 1.** Source data for *Figure 5B*.

**Source data 2.** Source data for *Figure 5C*.

**Source data 3.** Source data for *Figure 5D*.

aerobic respiration and flavin-based electron transfer, with the downstream portion of the pathway only being required for aerobic respiration (*Figure 5A*).

Based on the proposed roles of quinones in respiration, we hypothesized that the severe phenotypes previously described for the upstream quinone biosynthesis mutants were due to an imbalance in the NAD⁺/NADH ratio. To address this hypothesis, we first confirmed that the Δ*menB* strain, which is defective in upstream quinone biosynthesis, exhibited a phenotype similar to the Δ*QC/fmnB* strain for plaque formation and in the murine infection model (*Figure 5B and C*). We next tested the effect of NOX expression on virulence phenotypes for the Δ*menB* strain. NOX expression rescued Δ*menB* phenotypes for plaque formation and in the murine infection model to a strikingly similar extent as the

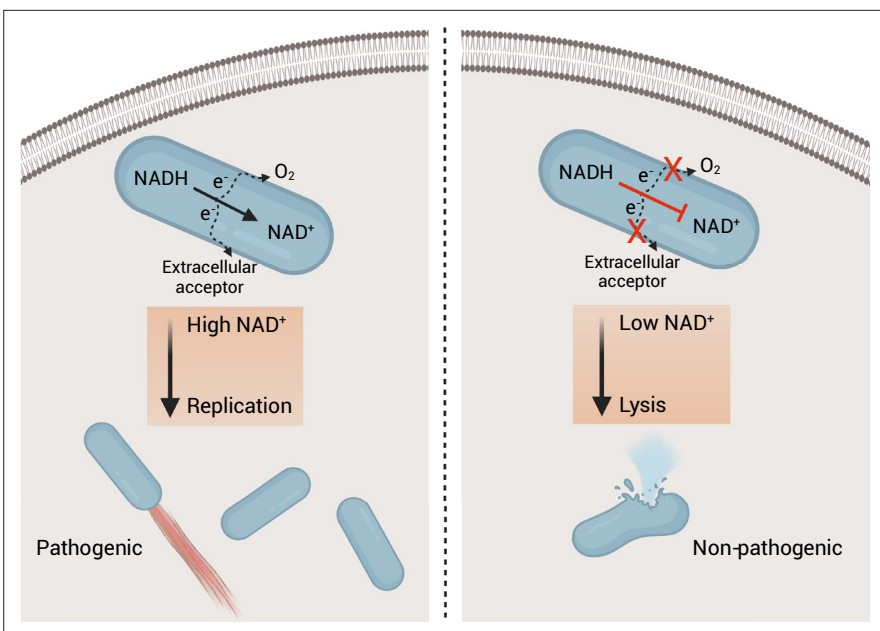

**Figure 6.** Model of the role of respiration in *L. monocytogenes* pathogenesis. On the left, an intracellular bacterium with the ability to oxidize NADH and transfer electrons through the aerobic and extracellular electron transfer electron transport chains can regenerate and maintain high NAD$^+$ levels allowing the bacterium to grow and be virulent. On the right, an intracellular bacterium unable to regenerate NAD$^+$, by lacking the electron transport chains, is avirulent because it lyses in the cytosol of infected cells.

$\Delta QC/fmnB$ strain (*Figure 5B and C*). These results thus provide evidence that quinone biosynthesis is essential for respiration and that the severity of the $\Delta menB$ phenotype is largely due to the role of respiration in regenerating NAD$^+$.

Numerous adaptations allow *L. monocytogenes* to colonize the host cytosol, including resistance to bacteriolysis. Minimizing bacteriolysis within the host cytosol is important to the pathogen because it can activate the host's innate immune responses, including pyroptosis, a form of programmed cell death, which severely reduces *L. monocytogenes* virulence (*Sauer et al., 2010*). *L. monocytogenes* strains deficient for the upstream quinone biosynthesis steps were previously identified as having an increased susceptibility to bacteriolysis in the macrophage cytosol (*Chen et al., 2017*). We thus hypothesized that decreased virulence of respiration-deficient strains might relate to increased cytosolic bacteriolysis.

Using a previously described luciferase-based assay to quantify cytosolic plasmid release, we confirmed that the $\Delta menB$ strain exhibited increased intracellular bacteriolysis (*Figure 5D*; *Sauer et al., 2010*). We further found that NOX expression rescued $\Delta menB$ bacteriolysis, but not a comparable bacteriolysis phenotype in a $\Delta glmR$ strain that was previously shown to result from unrelated deficiencies in cell wall biosynthesis (*Figure 5D*; *Pensinger et al., 2021*). These studies thus show that efficient NAD$^+$ regeneration is essential for limiting cytosolic bacteriolysis and suggest a model whereby respiration-mediated NAD$^+$ regeneration promotes virulence, in part, by maintaining cell viability and facilitating evasion of innate immunity (*Figure 6*).

## Discussion

Cellular respiration is one of the most fundamental aspects of bacterial metabolism and a validated antibiotic target. Despite its importance, the role of cellular respiration in systemic bacterial pathogenesis has remained largely unexplained. The studies reported here address the basis of respiration in the pathogen *L. monocytogenes*, identifying two electron transport chains that are partially functionally redundant and essential for pathogenesis. We find that restoring NAD$^+$ regeneration to respiration-deficient *L. monocytogenes* strains through the heterologous expression of NOX prevents bacteriolysis within the host cytosol and rescues pathogenesis. These findings thus support the

conclusion that NAD$^+$ regeneration represents a major role of *L. monocytogenes* respiration during pathogenesis.

Our results clarify several aspects of the basis and significance of energy metabolism in *L. monocytogenes*. In particular, our studies establish the relationship between *L. monocytogenes*' two electron transport chains – confirming previous observations that flavin-based electron transfer enhances anaerobic *L. monocytogenes* growth and revealing a novel aerobic function of this pathway (**Light et al., 2018**; **Zeng et al., 2021**). While the benefit of flavin-based electron transfer was only apparent in the absence of aerobic respiration, identifying the substrates and functions of aerobic activation of this pathway may provide an interesting avenue for future studies.

Our studies further reveal that *L. monocytogenes* employs a respiro-fermentative metabolic strategy characterized by production of the reduced fermentation products lactate and ethanol in the absence of an electron acceptor and acetate when a respiratory pathway is activated. This respiro-fermentative metabolism is consistent with the proton motive force being less central to *L. monocytogenes* energy metabolism and with a primary role of respiration being to unleash ATP production via acetate kinase catalyzed substrate-level phosphorylation (**Figure 1D**).

The importance of cellular respiration for non-proton motive force-related processes is further supported by observations about the ability of heterologous NOX overexpression to rescue the severe pathogenesis phenotypes of respiration-deficient *L. monocytogenes* strains. NOX expression fully rescued in vitro growth defects and partially rescued virulence in the mouse model of disease, suggesting that NAD$^+$ regeneration represents the sole function of respiration in some cell types and a major (but not sole) function of respiration in systemic disease. These findings suggest that a presently unaccounted for proton motive force-dependent aspect of microbial physiology is likely important for systemic disease. Considering the significance of cellular respiration as an antibiotic target, these insights into the role respiration be relevant for future drug development strategies.

While our studies provide evidence that NAD$^+$ regeneration is critical for preventing intracellular bacteriolysis, some ambiguity remains regarding the molecular mechanism linking NAD$^+$/NADH

**Table 1.** Bacterial strains used in this study.

| Strains | Strain number | Reference |
|---|---|---|
| *Listeria monocytogenes* (wildtype) | 10403S | **Bécavin et al., 2014** |
| Δ*cydAB*/Δ*qoxA* | DP-L6624 | **Chen et al., 2017** |
| Δ*cydAB*/Δ*qoxA*/*fmnB*::tn | DP-L7190 | This study |
| Δ*fmnB* | DP-L7195 | This study |
| Wildtype + pPL2 NOX | DP-L7188 | This study |
| Δ*cydAB*/Δ*qoxA* + pPL2 NOX | DP-L7189 | This study |
| Δ*cydAB*/Δ*qoxA*/*fmnB*::tn + pPL2 NOX | DP-L7191 | This study |
| Δ*flaA* | DP-L5986 | **Nguyen et al., 2020** |
| Δ*ndh1*/*ndh2*::tn | DP-L6626 | This study |
| Δ*ndh1*/*ndh2*::tn + pPL2 NOX | DP-L7253 | This study |
| Wildtype + pBHE573 | JDS18 | **Sauer et al., 2010** |
| Wildtype + pPL2 NOX + pBHE573 | JDS2328 | This study |
| Δ*menB* + pBHE573 | JDS1191 | **Chen et al., 2017** |
| Δ*menB* + pPL2 NOX + pBHE573 | JDS2333 | This study |
| Δ*hly* + pBHE573 | JDS19 | **Sauer et al., 2010** |
| Δ*glmR* + pBHE573 | JDS21 | **Sauer et al., 2010** |
| Δ*glmR* + pPL2 NOX + pBHE573 | JDS2329 | This study |
| *Escherichia coli* | SM10 | |
| pPL2-NOX | DP-E7206 | This study |
| pBHE573 | JDS17 | **Sauer et al., 2010** |

imbalance to the loss of *L. monocytogenes* virulence. One potential clue comes from a recent study of the transcriptional regulator Rex. Rex senses a low $NAD^+$/NADH ratio and derepresses reductive fermentation pathways, including those that produce lactate and ethanol, and a *L. monocytogenes* strain deficient in Rex exhibited decreased virulence (*Halsey et al., 2021*). Activation of part of the Rex regulon may at least partially account for the $NAD^+$/NADH-dependent phenotypes observed in our studies. The centrality of $NAD^+$ regeneration to *L. monocytogenes* also falls in line with relatively recent studies of mammalian respiration. Several studies have shown that the inability of respiration-deficient mammalian cells to regenerate $NAD^+$ impacts anabolic metabolisms and inhibits growth (*Birsoy et al., 2015*; *Li et al., 2020*; *Sullivan et al., 2015*; *Titov et al., 2016*). Our discovery of a similar role of respiration in a bacterial pathogen thus suggests that the importance of respiration for $NAD^+$ regeneration is a fundamental property conserved across the kingdoms of life.

## Materials and methods
### Bacterial culture and strains
All strains of *L. monocytogenes* used in this study were derived from the wildtype 10403S (streptomycin-resistant) strain (see *Table 1* for references and additional details). The *L. lactis* water-forming *nox* (NCBI accession WP_010905313.1) was cloned into the pPL2 vector downstream of the constitutive $P_{hyper}$ promoter and integrated into the *L. monocytogenes* genome via conjugation, as previously described (*Lauer et al., 2002*; *Shen and Higgins, 2005*). The Δ*QC*/*fmnB* strain was generated from Δ*QC* and *fmnB*::tn strains using generalized transduction protocols with phage U153, as previously described (*Hodgson, 2000*; *Reniere et al., 2016*).

L. monocytogenes cells were grown at 37°C in filter-sterilized BHI media. Growth curves were spectrophotometrically measured by optical density at a wavelength of 600 nm ($OD_{600}$). An anaerobic chamber (Coy Laboratory Products) with an environment of 2% $H_2$ balanced in $N_2$ was used for anaerobic experiments. Media was supplemented with 50 mM ferric ammonium citrate or 50 mM fumarate for experiments that addressed the effect of electron acceptors on *L. monocytogenes* growth.

### Plaque assays
*L. monocytogenes* strains were grown overnight slanted at 30°C and were diluted in sterile phosphate-buffered saline (PBS). Six-well plates containing $1.2 \times 10^6$ mouse L2 fibroblast cells per well were infected with the *L. monocytogenes* strains at a multiplicity of infection (MOI) of approximately 0.1. At 1-hour post-infection, the L2 cells were washed with PBS and overlaid with Dulbecco's Modified Eagle Medium (DMEM) containing 0.7% agarose and gentamicin (10 µg/mL) to kill extracellular bacteria, and then plates were incubated at 37°C with 5% $CO_2$. At 72-hour post-infection, L2 cells were overlaid with a staining mixture containing DMEM, 0.7% agarose, neutral red (Sigma), and gentamicin (10 µg/mL), and plaques were scanned and analyzed using ImageJ, as previously described (*Reniere et al., 2016*; *Sun et al., 1990*).

### Intracellular macrophage growth curves
*L. monocytogenes* strains were grown overnight slanted at 30°C and were diluted in sterile PBS. A total of $3 \times 10^6$ BMMs from C57BL/6 mice were seeded in 60 mm non-TC treated dishes containing 14 12 mm glass coverslips in each dish and infected at an MOI of 0.25 as previously described (*Portnoy et al., 1988*; *Reniere et al., 2016*).

### Mouse virulence experiments
*L. monocytogenes* strains were grown at 37°C with shaking at 200 r.p.m. to mid-logarithmic phase. Bacteria were collected and washed in PBS and resuspended at a concentration of $5 \times 10^5$ colony-forming units (CFU) per 200 µL of sterile PBS. The 8-week-old female CD-1 mice (Charles River) were then injected with $1 \times 10^5$ CFU via the tail vein. At 48 hours post-infection, spleens and livers were collected, homogenized, and plated to determine the number of CFU per organ.

### $NAD^+$/NADH assay
*L. monocytogenes* strains were grown at 37°C with shaking at 200 r.p.m. to mid-logarithmic phase. Cultures were centrifuged and then resuspended in PBS. Resuspended bacteria were then lysed by

vortexing with 0.1-mm-diameter zirconia–silica beads for 10 min. Lysates were used to measure $NAD^+$ and NADH levels using the NAD/NADH-Glo assay (Promega, G9071) by following the manufacturer's protocol.

### Fermentation product measurements

Organic acids and ethanol were measured by high-performance liquid chromatography (Agilent, 1260 Infinity), using a standard analytical system (Shimadzu, Kyoto, Japan) equipped with an Aminex Organic Acid Analysis column (Bio-Rad, HPX-87H 300 × 7.8 mm) heated at 60°C. The eluent was 5 mM of sulfuric acid, used at a flow rate of 0.6 mL/min. We used a refractive index detector 1260 Infinity II RID and a 1260 Infinity II variable wavelength detector. A five-point calibration curve based on peak area was generated and used to calculate concentrations in the unknown samples.

### Motility assay

*L. monocytogenes* strains were grown overnight slanted at 30°C and were diluted in sterile PBS. Cultures were normalized to an $OD_{600}$ of 1.0 and 1 µL of cultures were inoculated on semisolid BHI 0.3% agar. Mutant swarming diameters relative to wildtype were quantified following 48 hours incubation at 30°C.

### Intracellular bacteriolysis assay

Bacteriolysis assays were performed as previously described (*Chen et al., 2017*). Briefly, immortalized *Ifnar*$^{-/-}$ macrophages were plated at a concentration of $5 \times 10^5$ cells per well in a 24-well plate. Cultures of *L. monocytogenes* strains were grown overnight slanted at 30°C and diluted to a final concentration of $5 \times 10^8$ CFU per mL. Diluted cultures were then used to infect macrophages at an MOI of 10. At 1-hour post-infection, wells were aspirated, and the media was replaced with media containing 50 µg/mL gentamicin. At 6 hours post-infection, media was aspirated, and macrophages were lysed using TNT lysis buffer (20 mM Tris, 200 mM NaCl, 1% Triton [pH 8.0]). Lysate was then transferred to 96-well plates and assayed for luciferase activity by luminometry (Synergy HT; BioTek, Winooski, VT).

## Acknowledgements

Research reported in this publication was supported by funding from the National Institutes of Health (T32GM007215 to HBS, R01AI137070 to J-DS, R01AI073843 to EPS, 1P01AI063302, and 1R01AI27655 to DAP, and K22AI144031 to SHL), the National Academies of Sciences, Engineering, and Medicine (Ford Foundation Fellowship to RR-L), the University of California Dissertation-Year Fellowship (to RR-L), and the Searle Scholars Program (to SHL). VMRR holds a Postdoctoral Enrichment Program Award from the Burroughs Wellcome Fund and acknowledges support from the Academic Pathways Postdoctoral Fellowship at Vanderbilt University and the Howard Hughes Medical Institute Hanna H Gray Fellows Program. Work at the Molecular Foundry was supported by the Office of Science, Office of Basic Energy Sciences, of the U.S. Department of Energy under Contract No. DE-AC02-05CH11231.

## Additional information

### Competing interests

Denis V Titov: is a co-inventor on a filed patent describing the use of NOX. (US Patent App. 15/749,218). The other authors declare that no competing interests exist.

### Funding

| Funder | Grant reference number | Author |
| --- | --- | --- |
| National Institutes of Health | T32GM007215 | Hans B Smith |
| National Institutes of Health | R01AI137070 | John Demian Sauer |

| Funder | Grant reference number | Author |
|---|---|---|
| National Institutes of Health | R01AI073843 | Eric P Skaar |
| National Institutes of Health | 1P01AI063302 | Daniel A Portnoy |
| National Institutes of Health | 1R01AI27655 | Daniel A Portnoy |
| National Institutes of Health | K22AI144031 | Samuel H Light |
| National Academies of Sciences, Engineering, and Medicine | Ford Foundation Fellowship | Rafael Rivera-Lugo |
| University of California | Dissertation-Year Fellowship | Rafael Rivera-Lugo |
| Kinship Foundation | Searle Scholars Program | Samuel H Light |
| Howard Hughes Medical Institute | Hanna H. Gray Fellows Program | Valeria M Reyes Ruiz |
| Burroughs Wellcome Fund | Postdoctoral Enrichment Program | Valeria M Reyes Ruiz |
| Vanderbilt University | Academic Pathways Postdoctoral Fellowship | Valeria M Reyes Ruiz |
| Department of Energy | DE-AC02-05CH11231 | Sara Tejedor-Sanz Caroline M Ajo-Franklin |

The funders had no role in study design, data collection and interpretation, or the decision to submit the work for publication.

## Author contributions

Rafael Rivera-Lugo, Conceptualization, Investigation, Supervision, Writing – original draft; David Deng, Denis V Titov, Conceptualization, Investigation; Andrea Anaya-Sanchez, Eugene Tang, Valeria M Reyes Ruiz, Hans B Smith, John-Demian Sauer, Eric P Skaar, Caroline M Ajo-Franklin, Investigation; Sara Tejedor-Sanz, Daniel A Portnoy, Conceptualization, Investigation, Writing – original draft; Samuel H Light, Conceptualization, Funding acquisition, Investigation, Supervision, Writing – original draft

## Author ORCIDs

Rafael Rivera-Lugo http://orcid.org/0000-0002-2346-2297
Denis V Titov http://orcid.org/0000-0001-5677-0651
John-Demian Sauer http://orcid.org/0000-0001-9367-794X
Samuel H Light http://orcid.org/0000-0002-8074-1348

## Ethics

All animal work was performed in strict accordance with the recommendations in the Guide for the Care and Use of Laboratory Animals of the National Institutes of Health. Protocols were reviewed and approved by the Animal Care and Use Committee at the University of California, Berkeley (AUP 2016-05-8811).

## Decision letter and Author response

Decision letter https://doi.org/10.7554/eLife.75424.sa1
Author response https://doi.org/10.7554/eLife.75424.sa2

## Additional files

### Supplementary files
• Transparent reporting form

### Data availability
All data generated or analyzed during this study are included in the manuscript and supporting files.

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
