## [Editor Report]

In this study, authors report that a major role of respiration in *Listeria monocytogenes* pathogenicity, is to control redox balance (NAD^+^ regeneration) rather than generation of proton motive force. This is a new way of perceiving respiration that should be of interest to the microbiology community and broader readership.

---

## [Decision Letter]

**Decision letter after peer review:**

Thank you for submitting your article "*Listeria monocytogenes* requires cellular respiration for NAD^+^ regeneration and pathogenesis" for consideration by *eLife*. Your article has been reviewed by 3 peer reviewers, and the evaluation has been overseen by a Reviewing Editor and Gisela Storz as the Senior Editor. The following individuals involved in review of your submission have agreed to reveal their identity: Andreas Baumler (Reviewer #1); Andrez Vazquez-torres (Reviewer #3).

Essential revisions:

(1) Maintaining redox balance was essential in tissue culture assays (Figure 5A and B), but only partially accounted for attenuation of a respiration-deficient mutant during growth in the murine liver and spleen. Seeing the relatively modest effect on the recovery of virulence associated with the heterologous expression of NOX in the liver, oxidative phosphorylation may still play an important role during growth in this tissue. Thus, the claim in the abstract and elsewhere in the paper "that NAD+ regeneration, rather than oxidative phosphorylation, represents the primary role of *L. monocytogenes* respiration" should be tempered down and authors should consider roles for respiration other than redox balance.

(2) The use of the heterologous NOX system can rescue a series of defects, from acetic acid production, plaque assay and macrophage growth, virulence in murine infection model, to menB complementation including lysis. This suggests dispensability of pmf-dependent processes in all the phenotypes listed above. Please test whether well-known pmf-dependent processes (transport, protein secretion and or motility) remain altered in respiration deficient mutants (menI, ∆QC, ∆QC fmn) expressing the NAD-regenerating NOX system.

3) The primary effect of fmn or QC mutations is to enhance the amount of oxidized (D)MK, which in turn might well impact the pool of NADH. However, one cannot rule out other effects due to imbalanced reduced/oxidized (D)MK per se. Since NADH dehydrogenases are the effectors of NADH oxidation, please test single or double ∆ndh1, ∆ndh2 and/or ndh mutants. Does heterologous expression of NOX rescue the phenotypes of such mutants?

(4) Add statistics for comparisons of wildtype and NOX-complemented ∆QC, ∆QC/fmnB, and ∆menB strains (Figures 4c, 4d, and 5c).

---

## [Author Response]

Essential revisions:(1) Maintaining redox balance was essential in tissue culture assays (Figure 5A and B), but only partially accounted for attenuation of a respiration-deficient mutant during growth in the murine liver and spleen. Seeing the relatively modest effect on the recovery of virulence associated with the heterologous expression of NOX in the liver, oxidative phosphorylation may still play an important role during growth in this tissue. Thus, the claim in the abstract and elsewhere in the paper "that NAD+ regeneration, rather than oxidative phosphorylation, represents the primary role of *L. monocytogenes* respiration" should be tempered down and authors should consider roles for respiration other than redox balance.

Good point. We have re-worded the abstract and relevant part of the discussion to state that NAD+ regeneration is “a major role” of respiration.

(2) The use of the heterologous NOX system can rescue a series of defects, from acetic acid production, plaque assay and macrophage growth, virulence in murine infection model, to menB complementation including lysis. This suggests dispensability of pmf-dependent processes in all the phenotypes listed above. Please test whether well-known pmf-dependent processes (transport, protein secretion and or motility) remain altered in respiration deficient mutants (menI, ∆QC, ∆QC fmn) expressing the NAD-regenerating NOX system.

Thanks for the suggested control. We performed additional experiments which confirm that respiration-deficient mutants are less motile and that NOX expression does not impact this phenotype. These results are now presented in Figure 3—figure supplement 1.

3) The primary effect of fmn or QC mutations is to enhance the amount of oxidized (D)MK, which in turn might well impact the pool of NADH. However, one cannot rule out other effects due to imbalanced reduced/oxidized (D)MK per se. Since NADH dehydrogenases are the effectors of NADH oxidation, please test single or double ∆ndh1, ∆ndh2 and/or ndh mutants. Does heterologous expression of NOX rescue the phenotypes of such mutants?

Good point. The revised manuscript includes data in Figure 4A showing that the ∆ndh1/ndh2 mutant exhibits similar, NOX-rescuable, phenotypes in the plaque assay. The effect of NOX on this strain and the menB mutant (which lacks quinones) strongly suggests that NAD redox state and not is the primary driver of observed phenotypes.

4) Add statistics for comparisons of wildtype and NOX-complemented ∆QC, ∆QC/fmnB, and ∆menB strains (Figures 4c, 4d, and 5c).

Added.